# Prevalence and correlates of metabolic syndrome and its components in adults with psychotic disorders in Eldoret, Kenya

**Edith Kwobah**[1]*, **Nastassja Koen**[2,3,4], **Ann Mwangi**[5], **Lukoye Atwoli**[6‡], **Dan J. Stein**[2,3,4‡]

**1** Department of Mental Health, Moi Teaching and Referral Hospital, Eldoret, Kenya, **2** Department of Psychiatry and Mental Health, University of Cape Town, Cape Town, South Africa, **3** South African Medical Research Council [SAMRC], Unit on Risk and Resilience in Mental Disorders, Cape Town, South Africa, **4** Neuroscience Institute, University of Cape Town, Cape Town, South Africa, **5** Department of Behavioral Sciences, Moi University School of Medicine, Eldoret, Kenya, **6** Department of Mental Health, Moi University School of Medicine, Eldoret, Kenya

‡ These authors are joint senior authors on this work.
* eckamaru@gmail.com

**Data Availability Statement:** All relevant data are within the manuscript and its Supporting Information files.

**Funding:** The PhD thesis from which this paper was developed was supported by the

## Abstract

### Background

A high prevalence of metabolic syndrome and its components in patients with psychotic disorders may increase the risk for cardiovascular diseases. Unfortunately, relatively little work in this field has emerged from low-resourced contexts. This study investigated the prevalence, correlates, and treatment patterns of metabolic disorders in patients with psychotic disorders in Western Kenya.

### Methods

300 patients with psychosis and 300 controls were recruited at Moi Teaching and Referral Hospital in Eldoret, Kenya. Data on demographic characteristics, weight, height, abdominal circumference, blood pressure, blood glucose, lipid profile, and treatments were collected. Categorical and continuous data were compared between the patient and control groups using Pearson's chi-squared tests and t-tests, respectively. Variables found to be significantly different between these groups were included in logistic regression models to determine potential predictors of metabolic syndrome.

### Results

Compared to controls, patients with psychosis were found to have a higher mean random blood glucose [5.23 vs 4.79, p = 0.003], higher body mass index [5.23 vs 4.79, p = 0.001], higher triglycerides [1.98 vs 1.56, p<0.001], larger waist circumference [89.23 vs 86.39, p = 0.009] and lower high density lipoprotein [1.22 vs 1.32, p<0.001]. The odds of developing metabolic syndrome were increased with age [OR = 1.05, CI: 1.02–1.07] and presence of a psychotic disorder [OR = 2.09 [CI 1.23–3.55]; and were reduced with female gender [OR 0.41, CI 0.25–0.67], among those who were never married [OR 0.52, CI 0.28–0.94] and among the widowed/separated/ divorced marital status [OR 0.38, CI 0.17–0.81]. While the

Neuropsychiatric Genetics of African Populations (NeuroGAP) - Psychosis study hosted at the Stanley Global at the Broad Institute and the Harvard T.H Chan School of public health, USA. The funders had no role in study design, data collection and analysis, decision to publish, or preparation of the manuscript.

**Competing interests:** The authors declare no conflicting interest.

majority of patients received treatment with olanzapine, there was no association between olanzapine use and metabolic syndrome and its components. More than half of the patients in this study sample were not receiving treatment for the various components of metabolic syndrome.

## Conclusion

In the study setting of Eldoret, metabolic syndrome and its components were more prevalent among patients with psychotic disorders than in controls; and a clear treatment gap for these disorders was evident. There is a need for efforts to ensure adequate screening and treatment for these physical disorders in resource-limited settings.

## Introduction

Cardiovascular disorders [CVDs] are the leading cause of death worldwide [1]; and there is evidence to suggest an increased risk among patients with psychosis compared to the general population [2, 3]. Metabolic syndrome comprises a cluster of conditions, including diabetes mellitus, hypertension, dyslipidemia and obesity; and may increase CVD risk through a number of pathophysiological mechanisms [4]. In patients with psychotic disorders, the prevalence of metabolic syndrome is estimated at 30% [5], diabetes at 10% [6], and hypertension at 20% [7]. This high burden of co-morbidity of psychotic disorders and metabolic syndrome and its components has been found to contribute significantly to the excess mortality in this population [8, 9]; and has partly been attributed to the use of antipsychotics [primarily clozapine and olanzapine] [10]. The risk of CVD-related mortality and morbidity is further complicated by current gaps in screening, diagnosis and treatment for cardiovascular risk amongst patients with psychotic disorders [11, 12].

While there is substantial literature on the prevalence and determinants of metabolic syndrome in psychotic disorders in the developed world, low- and middle-income countries [LMICs]–and sub-Saharan Africa [SSA] in particular–are largely under-represented [13]. Within the limited data emerging from SSA, certain countries [such as South Africa] are notably over-represented [14]. The need for further literature in resource-limited settings is highlighted by compelling evidence from the World Health Organization [WHO] that approximately 75% of all reported CVD-related deaths in the general population take place in LMICs [15]. To the best of our knowledge, there is no published evidence from Kenya on the prevalence, correlates, and treatment of metabolic syndrome in patients with psychotic disorders.

Moi Teaching and Referral Hospital [MTRH] in Eldoret is the second largest national referral hospital in Kenya, and is representative of many academic centers of Psychiatry in East Africa. The 70-bed inpatient unit for psychotic disorders serves a catchment area of more than 15 million people from both rural and urban settings, including diverse ethnic groups; and is a center for the training of medical students and other healthcare workers. In order to address the current gap in the literature, this study aimed to assess the prevalence, correlates, and treatment of metabolic syndrome and its components among patients with psychotic disorders in this setting.

## Methods

We undertook a cross-sectional descriptive survey nested within the Neuropsychiatric Genetics of African Populations [NeuroGAP] initiative, an ongoing, multisite, case control study

aiming to explore potential genetic underpinnings of psychotic disorders in African populations [16, 17].

Patients [cases] were eligible for inclusion if they were [i] aged 18 years or older; [ii] had been followed up in the MTRH outpatients' service for a minimum of 6 months; and [iii] had a diagnosis [as made by a psychiatrist] of either schizophrenia, schizoaffective disorder or bipolar mood disorder with or without psychotic features [together referred to as psychotic disorders for our purposes]. Psychiatry diagnoses were made using the Mini International Neuropsychiatric Interview [MINI], for the Diagnostic Statistical Manual Version 5 [DSM-5] [18]. Pregnant women were excluded from participation, due to evidence that substantial CVD-related changes may occur during pregnancy [19]. Individuals who did not have the capacity to consent—as assessed by the University of California, San Diego Brief Assessment of Capacity to Consent [UBACC]–were also excluded [20, 21]. Controls were adults aged 18 years and older without a mental illness, who were not receiving any psychotropic medication, and who were not being treated for acute alcohol or drug intoxication. The controls were drawn from hospital visitors, patients attending other clinics, and hospital staff.

Demographic data including age, sex, marital status and highest level of education were collected. Clinical measures of weight, height, abdominal circumference and blood pressure were also taken; and blood drawn for random blood glucose and non- fasting lipid profile testing. A non-fasting assessment was undertaken due to anticipated difficulties in ensuring or ascertaining a fasted state among mentally ill participants. There is evidence to suggest that, in assessing for cardiovascular risk, non- fasting lipid profile is adequate, time-efficient, and patient-friendly. A fasting test is thus recommended only for those with triglycerides > 5 mmol/L [22]. In the current study, the Expert Panel on Detection, Evaluation, and Treatment of High Blood Cholesterol in Adults [Adult Treatment Panel III] was used to define Metabolic Syndrome [MS] [23], adapted to include the random blood glucose equivalent instead of fasting glucose. Patient records were reviewed to establish documentation of treatment for metabolic syndrome and its component conditions. This was complemented by patients' self-report of other medication use.

Data were entered into REDcap, [Research Electronic Data Capture] a secure, web-based software platform designed to support data capture for research studies [24], cleaned and then exported into Stata version 15 for analysis [Statistical Software: Release 14. College Station, TX: StataCorp LP]. Pearson chi-squared tests, [or Fisher's exact tests where applicable], were used to assess associations between categorical variables, while t-tests [or Wilcoxon rank sum test where applicable] were used to compare means of continuous variables. Normality assumption was assessed using the Shapiro Wil test. When comparing the various components of MS between patients and controls, to account for multiple testing, we adjusted the significance level to $0.05/9 = 0.005$. In all other analysis a p-value less than 0.05 was considered to be statistically significant. Variables that were significant in bivariate analysis were included in the multivariate logistic regression model to explore potential associations with metabolic syndrome.

This study was approved by the MTRH/Moi University School of Medicine Institutional Research and Ethics Committee [IREC/2017/90] and by the Human Research Ethics Committee of the Faculty of Health Sciences, University of Cape Town [HREC/286/2017]. All participants provided written informed consent prior to participation.

## Results

### Sociodemographic characteristics

Data were collected between July 2018 and March 2019; 300 patients with psychosis and 300 controls were recruited. Three patients were excluded from analyses—two were found to have epilepsy-induced psychosis on further evaluation; and one had substance-induced psychosis.

The study participants were generally young, with a mean age of 33 years among patients [SD 26, 40] and the age range of 18 to 67 years, while for controls the mean age was 35 years among [SD 26.5, 41] with an age range of 18 to 69 years. Statistically significant differences in marital status, education and occupation were evident between patients and controls [**Table 1**]. Specifically, patients were less likely to be married, less likely to have tertiary education, and more likely to be unemployed.

## Prevalence of metabolic syndrome and its components

Compared to controls, a higher proportion of patients had metabolic syndrome, elevated triglycerides [47.1 vs 30.3, p<0.001], elevated total cholesterol [23.6 vs 14.7, p = 0.006], and obesity [45.1 vs 35.6, p = 0.037] (Fig 1).

Patients were also found to have a higher median random blood sugar, BMI and triglycerides and lower HDL, **Table 2**.

## Factors associated with metabolic syndrome

Compared to controls, patients were more likely to have metabolic syndrome [28.6 vs 19, p = 0.007]. In the adjusted logistic regression analysis, psychosis and increased age each were associated with metabolic syndrome [**Table 3**]. Factors found to be associated with decreased odds of metabolic syndrome were female gender, being widowed/separated/divorced, and never having been married.

## Factors associated with obesity as indicated by abnormal BMI

Compared to controls, a higher proportion of patients were found to be obese [45.1% vs 35.6%, p = 0.037]. In the adjusted logistic regression model, factors associated with increased odds of obesity were psychosis, age, and female gender. Having never been married [or being windowed/ separated/ divorced] was associated with decreased odds of obesity, **Table 4**

**Table 1. Sociodemographic characteristics of the study participants.**

| Variable | Patients | Controls | p-value |
|---|---|---|---|
| | N [%] | N [%] | |
| **Sex** | | | 0.238[1] |
| Male | 139 [46.8] | 126 [42.0] | |
| Female | 158 [53.2] | 174 [58.0] | |
| **Marital status** | | | **<0.001**[1] |
| Married | 103 [34.7] | 166 [55.3] | |
| Never married | 139 [46.8] | 100 [33.3] | |
| Widow/separated/divorced | 55 [18.5] | 34 [11.3] | |
| **Highest level of education** | | | **<0.001** |
| None | 23 [7.7] | 12 [4] | |
| Primary | 106 [35.7] | 41 [13.7] | |
| Secondary | 92 [31] | 80 [26.7] | |
| Tertiary | 76 [25.6] | 167 [55.7] | |
| **Occupation** | | | **<0.001**[1] |
| Unemployed | 143 [48.1] | 74 [24.7] | |
| Formal | 43 [14.5] | 144 [48] | |
| Self | 11[17.4] | 82 [27.3] | |

1 Chi-squared test 2 Fishers' exact test 3 Wilcoxon rank sum test.

### Psychotropic medication patterns and metabolic disorders

At the time of assessment, the vast majority [94.3%] of patients were being treated with an antipsychotic medication: 257/286 [89%] with olanzapine, 4 [1%] with chlorpromazine,

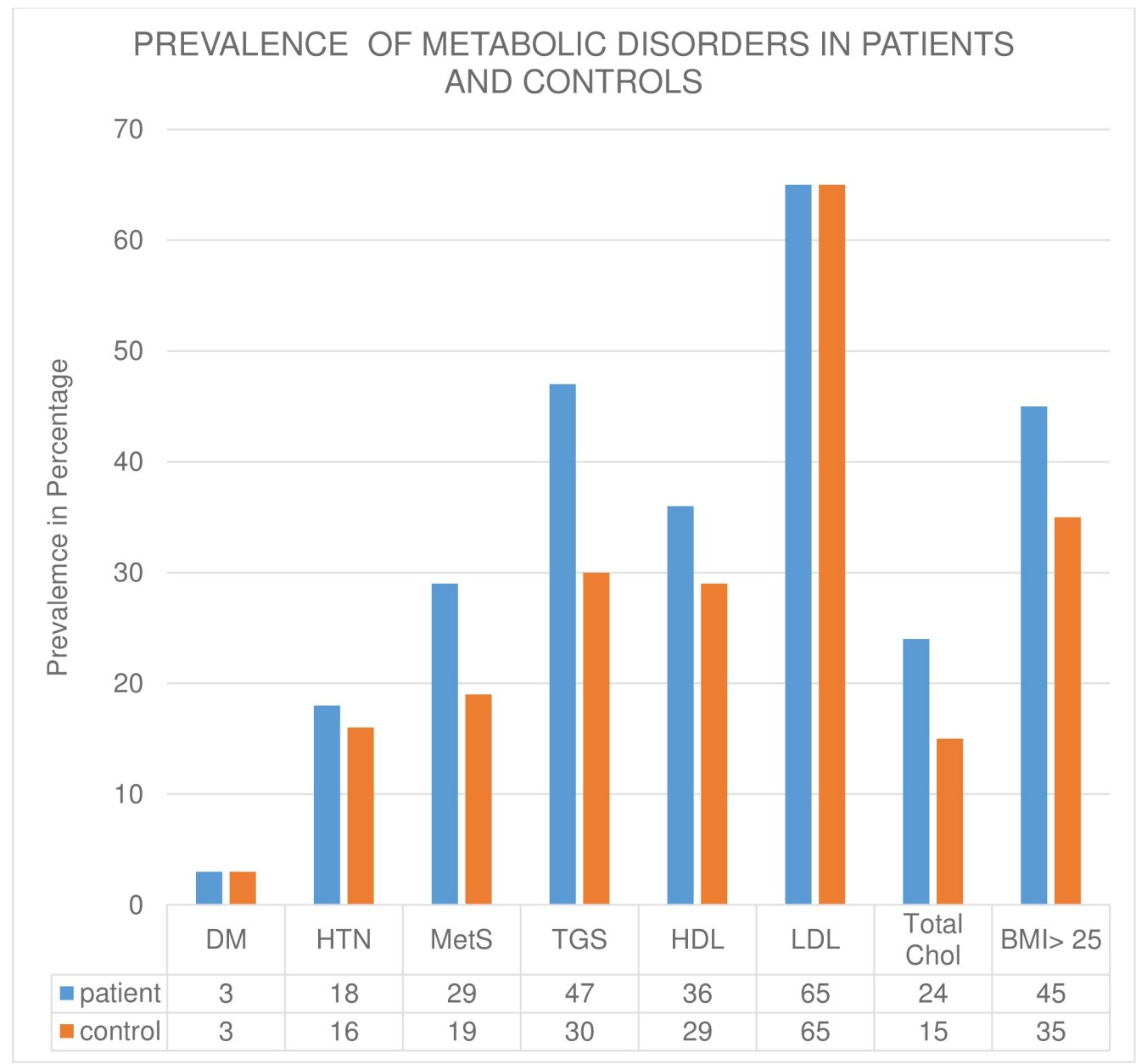

*.Met S - Metabolic Syndrome HTN- Hypertension BMI -Abnormal Body mass index    LDL - Low*

*High density Lipoproteins        HDL - High density Lipoproteins TGS - Triglycerides*

*Cho l- Cholesterol*

**Fig 1. Prevalence of metabolic disorders in patients and controls.**

**Table 2. A comparison of the median metabolic parameters between patients and controls.**

| Variable | Total | Patients | Controls | Wilcoxon Sum Rank test p-value* |
|---|---|---|---|---|
| | Median [IQR] | Median [IQR] | Median [IQR] | |
| Systolic BP | 118 [110, 128] | 119 [110, 128.5] | 118 [110, 128] | 0.2116 |
| Diastolic BP | 75.5 [68.5, 82.5] | 75.5 [69, 82] | 75.5 [68.5, 82.5] | 0.8329 |
| **RBS[1]** | **4.715 [4.22,5.27]** | **4.88 [4.31,5.5]** | **4.72 [4.22,5.27]** | **<0.0001** |
| **BMI[2]** | **24.13 [20.98,27.41]** | **24.57 [21.82,28.03]** | **24.13 [20.98,27.41]** | **0.0005** |
| **HDL[3]** | **1.19 [0.98,1.44]** | **1.12 [0.95,1.38]** | **1.19 [0.98,1.44]** | **<0.0001** |
| LDL[4] | 2.85 [2.37,3.53] | 2.86 [2.35,3.58] | 2.85 [2.37,3.53] | 0.5743 |
| Chol[5] | 4.31 [3.73,5.02] | 4.35 [3.69,5.13] | 4.31 [3.73,5.02] | 0.278 |
| **TG[6]** | **1.38[0.96,2.13]** | **1.57 [1.05,2.47]** | **1.38 [0.96,2.13]** | **<0.0001** |
| Waist circum[7] | 87 [79,95] | 88 [81,97.25] | 87[79,95] | 0.007 |

[1]Random blood sugar

[2]Body mass index

[3]High density lipoprotein

[4] Low Density Lipoprotein

[5]Cholesterol

[6]Triglycerides

[7] Waist circumference.

*Level of significance was adjusted to 0.005 to allow for multiple comparison.

18 [6%] with haloperidol and 1 [0.3%] with zuclopenthixol. Six participants were being treated with antidepressant medications. Neither olanzapine use, nor olanzapine dosage was significantly associated with an increased risk of metabolic syndrome, and an inverse relationship between olanzapine use and hypertension [29.9% vs 32.5%, p = 0.029] was noted.

## Treatment for components of metabolic syndrome

More than half [60%] of patients, and almost a quarter [22%] of controls were found to have undiagnosed diabetes mellitus, **Table 5**. The prevalence of untreated hypertension was also high–both in cases [65%] and in controls [47%]. We found no record of treatment with lipid-lowering drugs; or of any other medication to reduce the adverse cardiovascular effects of anti-psychotic treatment in this study sample.

**Table 3. Factors associated with the presence of metabolic syndrome.**

| Variable | Adjusted Odds Ratio | P value | 95% Conf Interval | |
|---|---|---|---|---|
| No psychosis | 1 | | | |
| **Psychosis** | **2.09** | **0.006** | **1.23** | **3.55** |
| **Age in years** | **1.05** | **<0.001** | **1.02** | **1.07** |
| Male | 1.00 | | | |
| **Female** | **0.41** | **<0.001** | **0.25** | **0.67** |
| Married | 1.00 | | | |
| **Never married** | **0.52** | **0.031** | **0.28** | **0.94** |
| **Widow/separated/divorced** | **0.38** | **0.013** | **0.17** | **0.81** |
| No education | 1.00 | | | |
| Primary | 0.48 | 0.112 | 0.19 | 1.19 |
| Secondary | 0.43 | 0.071 | 0.17 | 1.07 |
| Tertiary | 0.60 | 0.270 | 0.24 | 1.48 |

**Table 4. Factors associated with obesity as indicated by an elevated BMI.**

| Variable | Odds Ratio | P-value | [95% Conf. Interval | |
|---|---|---|---|---|
| No psychosis | 1 | | | |
| **Psychosis** | **2.30** | **<0.001** | **1.55** | **3.43** |
| **Age in years** | **1.05** | **<0.001** | **1.02** | **1.07** |
| Male | 1 | | | |
| **Female** | **2.67** | **<0.001** | **1.82** | **3.92** |
| Married | 1 | | | |
| Never married | 0.52 | 0.005 | 0.33 | 0.82 |
| **Widow/separated/divorced** | **0.44** | **0.003** | **0.26** | **0.75** |
| No education | 1 | | | |
| Primary | 0.86 | 0.703 | 0.39 | 1.90 |
| Secondary | 0.91 | 0.807 | 0.41 | 2.00 |
| Tertiary | 1.43 | 0.374 | 0.65 | 3.14 |

## Discussion

In this study we found; 1] A high prevalence of DM, HTN, dyslipidemia, obesity and metabolic syndrome in patients with psychosis,; [2] A particular risk profile for metabolic syndrome; and [3] gaps in documented treatment for patients with diabetes, hypertension and dyslipidemia.

### Prevalence of metabolic syndrome and its components

In this study we found that 28.6% of the patients had metabolic syndrome. These findings are comparable to an earlier South African study done among 276 patients with severe mental illness which reported a point prevalence of 23% [25]. This study was comparable to our study in terms of having a young study population [mean age of 34.7 compared to our mean age of 33 years], but differed in that the majority of participants were on first generation antipsychotics. Our findings are also comparable to a cross sectional study done in Uganda in 2019 among 309 patients with severe mental illness; this reported a point prevalence of 24% in a study sample that was again comparable to ours in terms of age [mean age was 38.5], and that again differed in that the majority of the patients were on first generation antipsychotics [26]. In the current study metabolic syndrome was associated with psychosis and increasing age. Of note

**Table 5. Prevalence of treated and untreated metabolic disorders.**

| Variable | Patients | Controls | P-value |
|---|---|---|---|
| | N [%] | N [%] | |
| **Diabetes Mellitus** | | | 0.115 |
| | 4 [40%] | 7 [78%] | |
| Treated[1] | | | |
| Untreated[2] | 6 [60%] | 2 [22%] | |
| **Total** | 10 [100%] | 9 [100%] | |
| **Hypertension** | | | 0.069 |
| Treated | 19 [35%] | 25 [53%] | |
| Untreated | 35 [65%] | 22 [47%] | |
| **Total** | 54 [100%] | 47 [100] | |

[1]Treated—evidence of medication in the patient's records.

[2]Untreated—no documented medication or intervention in the patient's records.

is that males were more likely to have metabolic syndrome than females; findings that differ from those reported in a South African study where the prevalence of metabolic syndrome among black African women were more than three times more compared to men in the same setting [The South African study was similar to the Kenyan study in terms of age—both a mean age less than 40 in both and obesity rates around 45%] [27]. The lower rates of metabolic syndrome among females in our setting is may be attributed to other factors such as genetics, and environmental factors [28, 29] which may be difficult to explain in entirety at this point but warrant further investigation.

This study found that [45%] of our study sample were obese as defined by elevated BMI. This is higher than the findings of a study done in Tunisia among 130 patients on management for bipolar mood disorder [mean age 37.9] which reported a point prevalence rate of 33.8% [30]. Similarly, a study done in South Africa in 84 patients, the majority of whom were on first generation antipsychotics reported a point prevalence of 24% [31]. It is however lower than found in a study done in Australia [n = 189, mean age was 39.4, and the majority were on atypical antipsychotics] which reported that 78% of the participants, were either overweight or obese [32]. In the current study, the prevalence of elevated BMI was significantly higher among patients than controls, which is similar to a range of studies undertaken in high income settings [33]. The current study established that elevated BMI was positively associated with female sex. This is similar to findings of the 2010 WHO Global Infobase which reported that in 87% of the 151 countries involved, prevalence of obesity in females exceeded that of males [34]. Though not explored in our study, gender differences in obesity may have its basis on genetic and hormonal factors which result in women having increased fat mass proportional to their body weight and increased subcutaneous adipose tissue, while men tend to have greater proportional lean mass [35, 36]. In addition to high rates of elevated BMI, the study also established a high prevalence of central obesity as indicated by abnormal waist circumference [35%]. This has significance as abdominal obesity is believed to have the highest impact on metabolic profile amongst Africans [37]. These findings indicate the need to screen for obesity, and the need to put interventions in place to lower these rates in order to promote cardiovascular health in LMIC settings.

This study found a high prevalence of dyslipidemia [65% had elevated LDL, 47% had elevated triglycerides and 36% had low HDL] inpatients with psychosis, with both cholesterol and triglyceride levels significantly higher in patients compared to controls. The excess of dyslipidemia among patients with psychosis compared to the general population is further supported by the findings of the Health and Demographic Surveillance System [HDSS] site in the general population in Western Kenya which reported that only 15% of the population had elevated plasma lipids [38]. They however are comparable to findings of a Saudi Arabian cross sectional study that included 992 patients with mental illness which reported that 32.8% had elevated triglycerides and 52.5% had low HDL cholesterol [39]. The high prevalence of dyslipidemia in this study is a clear indicator for the need for continuous monitoring and management of lipids in patients with psychosis in our setting in order to reduce the associated risk of CVD.

This study found that 18.2% of the patients with psychosis had elevated blood pressure, which is lower than a prevalence of 24% in the general population that was reported by the national survey in Kenya in 2015 [40]. This is contrary to other evidence which suggests that the rates of hypertension in patients with psychosis are higher than the general population perhaps in part due to several shared etiological pathways between hypertension and mental illness [41]. For example, a nationwide survey in Japan in patients with schizophrenia found a prevalence of hypertension of 30% [42], and a multisite study done in France reported a prevalence of 43% [43]. The lower prevalence of hypertension in the current study might be

accounted for by the relatively younger age of the participants. Current evidence suggests that blood pressure changes by 20 mmHg systolic and 10 mm Hg diastolic increment from age 30 to 65 years [44], which is attributed to age related changes at the cellular level, including alterations in the second messengers calcium and magnesium that affect the tone of the small arteries [45], as well as structural changes especially large artery stiffness [46]. These findings indicate a need for more studies among older patients with psychosis who might be more at risk of hypertension than the young population that this study engaged.

The prevalence of DM in our study sample [3.4%] is lower than similar studies done in Canada, 10%, n = 28,755] [47] and Singapore.19.1%.[48]. It is however comparable to the findings of the National Steps survey on NCD risk factors that was carried out in Kenya in 2015 among 4069 participants aged between 18 and 69 years, which reported a 2.4% prevalence of diabetes [49]. This is also within the range that was reported by one review that included 143 articles which indicated that the prevalence of diabetes in patients with psychotic disorders ranges widely, from 1·26% to 50% across studies [median 13%] [50]. One possible contribution to the lower prevalence in the current study sample could be the use of random blood sugar, which may then include some patients in a fasting state. These findings point to the need for better assessments of presence of DM, for example by use of HB1Ac, before concluding definitively that patients with psychosis in this setting have a low prevalence of DM.

## Antipsychotic use and metabolic disorders

The vast majority of patients in our study sample were being treated with olanzapine–this is likely due to the ready availability of this agent in our study setting, where olanzapine has been donated to the hospital, making it the most prescribed drug in the setting. Despite its near-ubiquitous use, we found no association between either use or dosing of olanzapine and the metabolic syndrome and its components, which was quiet unexpected and this could not be explained by any confounders as there were no demographic or clinical differences between those who were on olanzapine and those were not on it. This is surprising and in contrast to established evidence of the metabolic adverse effects of this agent. For example, a recent meta-analysis of 307studies reported that the highest metabolic side effects were associated with olanzapine and clozapine [51]. This study is different from the other African studies in South Africa [52, 31], Uganda [53] and Tunisia [30] which indicate that most of the patients in African settings are on first generation antipsychotics. The lack of association of olanzapine and metabolic syndrome or any of its components is quite unexpected given that olanzapine is one of the antipsychotics highly associated with metabolic side effects [54, 55]. The discrepancy may reflect insufficient power [as so few patients were not on olanzapine] or may be due to possible differences in metabolism and side-effect profile which may have a basis in varying genetic and environmental factors [56].

## Treatment for components of metabolic syndrome

In this study a majority of the patients were not on treatment for metabolic disorders, indicating a treatment gap. This is likely because other than routine measuring of body weight and blood pressure there is no formal protocol for monitoring key indicators of metabolic syndrome such as measurements of abdominal circumference, blood sugar, and lipid profile. These findings are in line with a South African study of 331 patients with psychosis, which reported very low intervention rates; 0.6% for metabolic syndrome;, 3.9% for abdominal obesity, 3.9% for hyperglycemia and 1.8% for dyslipidemia [57]. Similarly a study conducted in the United States of America s reported that diabetes care and hypertension control was 14–49% lower in patients with mental illness compared to unaffected controls [58]. Sub-optimal

care for cardiovascular risk factors in patients with mental illness has been linked to stigma; de-prioritisation of psychiatric [versus biomedical] symptoms; and patients' inability to describe their symptoms due to their mental illness [59]. Systemic challenges such as lack of capacity [in terms of knowledge and infrastructure] for monitoring the physical illness; and limited time and human resources may also contribute to the treatment gap for CVD risk factors [60]. For example, the "Lancet Commission for Protecting Physical Health in People with Mental Illness" reported a 1.4 to 2 increased risk of several CVD risk factors in individuals with mental illness and that people with mental illness were more likely to receive lower quality of care compared to the general population. This led to the commission recommending a multidisciplinary approach for addressing physical disorders among mentally ill patients [61]. Importantly, in our setting there are relatively few mental healthcare providers, and consequently a high workload, which would likely contribute to the suboptimal monitoring of physical conditions among mentally ill patients [62]. In future there is need to consider increasing human resources for mental health care [by increasing the number of available staff to provide care such as psychiatrists, psychiatry nurses, social workers, occupational therapists and psychologists], as well as creating community based mental health services to allow for closer monitoring including monitoring for CVD risk factors. Such monitoring seems particularly appropriate in settings that are reliant on donations of second generation antipsychotics.

## Limitations of the study

While our study presents novel and clinically relevant preliminary data on metabolic disorder and its components in patients with psychosis in Western Kenya, a number of methodological limitations should be borne in mind. First, as an observational study design was employed, we are unable to make causal claims about the associations found here. Second, convenience sampling could have resulted in selection bias. However, we see no reason to suspect that any particular subgroup of patients or controls was omitted. Third, the use of non—fasted glucose and lipids measurements among the patients may have resulted in an underestimation of diabetes mellitus in the study sample. In future, glycated hemoglobin [HbA1c] measurement could be undertaken to get more accurate assessments of the prevalence of DM in the study setting Fourth majority of the patients were on olanzapine which limited the ability to establish the role of various antipsychotics in increasing the risk of metabolic syndrome and its components in this setting. Finally, the use of records to establish treatments is limited by the documentation itself; our study findings might thus be reflective of incomplete records, rather than a true treatment gap. However, patient reports were consistent with hospital records.

## Conclusion

In our Western Kenya study setting, patients with psychosis exhibited a higher prevalence of metabolic syndrome and its component when compared to controls. A number of gaps in current practice–in terms of underdiagnoses and under treatment of these conditions were identified. In future, routine screening for various CVD risk factors could improve early identification of high-risk patients. Thereafter, coordinated and integrated interventions, drawing on multidisciplinary approaches and guided by contextually appropriate standardized guidelines, could be effective in improving treatment outcome and overall quality of life of patients with psychosis.

## Supporting information

**S1 Data.**
(ZIP)

## Acknowledgments

1. Professor Carl Lombard of the Biostatistics Unit at the South African Medical Research Council for his assistance with sample size calculation.

2. Dr Kamano, Internal medicine specialist, Moi University, for her insights as I developed the proposal.

3. The NeuroGAP Moi team Stella Gichuru, Eunice Jeptanui, Fredrick Ochieng and Ndenga Indagala, and to my research assistant Julius Barasa, for their support in data collection and other logistics.

4. I am indebted to the AMPATH Kenya and my employer, Moi teaching and referral hospital headed by Dr Wilson Aruasa for allowing me to use the facilities that were required to complete this study.

## Author Contributions

**Conceptualization:** Edith Kwobah, Nastassja Koen, Ann Mwangi, Lukoye Atwoli, Dan J. Stein.

**Data curation:** Edith Kwobah, Ann Mwangi, Dan J. Stein.

**Formal analysis:** Edith Kwobah, Ann Mwangi.

**Investigation:** Edith Kwobah.

**Methodology:** Edith Kwobah, Nastassja Koen, Ann Mwangi, Lukoye Atwoli, Dan J. Stein.

**Project administration:** Edith Kwobah.

**Resources:** Edith Kwobah, Lukoye Atwoli, Dan J. Stein.

**Supervision:** Nastassja Koen, Lukoye Atwoli, Dan J. Stein.

**Visualization:** Edith Kwobah.

**Writing – original draft:** Edith Kwobah.

**Writing – review & editing:** Edith Kwobah, Nastassja Koen, Ann Mwangi, Lukoye Atwoli, Dan J. Stein.

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
