## [Decision Letter · Decision Letter 0]

8 Sep 2020

PONE-D-20-18075

Prevalence and Correlates of Metabolic Syndrome and its Components in Adults with Psychotic Disorders in Eldoret, Kenya

PLOS ONE

Dear Dr. Kwobah,

Thank you for submitting your manuscript to PLOS ONE. After careful consideration, we feel that it has merit but does not fully meet PLOS ONE’s publication criteria as it currently stands. Therefore, we invite you to submit a revised version of the manuscript that addresses the points raised during the review process.

We look forward to receiving your revised manuscript.

Kind regards,

Alberto Milan

Academic Editor

PLOS ONE

Journal Requirements:

2. Please provide additional details regarding participant consent. In the Methods section, please ensure that you have specified (a) whether consent was informed and (b) what type you obtained (for instance, written or verbal). If your study included minors, state whether you obtained consent from parents or guardians. If the need for consent was waived by the ethics committee, please include this information

4. Please ensure that you refer to Figure 1 in your text as, if accepted, production will need this reference to link the reader to the figure.

Reviewers' comments:

Reviewer's Responses to Questions

**Comments to the Author**

1. Is the manuscript technically sound, and do the data support the conclusions?

Reviewer #1: Yes

Reviewer #2: Yes

2. Has the statistical analysis been performed appropriately and rigorously? 

Reviewer #1: Yes

Reviewer #2: Yes

3. Have the authors made all data underlying the findings in their manuscript fully available?

Reviewer #1: Yes

Reviewer #2: Yes

4. Is the manuscript presented in an intelligible fashion and written in standard English?

Reviewer #1: Yes

Reviewer #2: No

5. Review Comments to the Author

Reviewer #1: This article analyses the correlation of metabolic syndrome and psychotic disorders in adults with a diagnosis of schizophrenia, schizoaffective disorder or bipolar mood disorder, compared to healthy individuals in the Moi Teaching and Referral Hospital in Eldoret, Kenya.

It is an interesting paper which focuses on a low resource context, which is often overlooked. It shows a significant increase of certain metabolic syndrome parameters in patients vs. controls, such as HDL levels, waist circumference and TG.

It is an observational study with definite limitations, that are underlined in the article itself: HbA1c was not measured, and most patients were on the same antipsychotic drug olanzapine. Moreover, most patients and controls were relatively young (mean age 33 vs. 35), which generally lowers the rate of CV events and metabolic syndrome. The controls, however, appear to have been chosen randomly in an appropriate manner. The authors reviewed a number of sociodemographic and laboratory variables.

I think this article has dignity to be published by itself. It is also a good starting point for possible future studies, possibly with a greater number of subjects and a more varied medication history and additional information, such as HbA1c value, arterial wall stiffness studies, long-term follow-up of patients and controls.

Reviewer #2: This is a cross-sectional study of metabolic health in 300 patients with psychosis from a hospital setting in Western Kenya. In a low-resource environment, the study authors found that patients with psychosis were more likely to have metabolic dysregulation (higher fasting glucose, BMI, lipid levels, waist circumference). They also identified relevant factors (older age, male gender, married status) associated with metabolic syndrome. This study presents important clinical findings from a region that has been understudied in the literature.

1. Methods (Page 5)– please clarify whether the inclusion criteria for the study required psychotic symptoms (for example, bipolar disorder with psychotic features)

2. Methods (Page 6) – please add details on whether the blood glucose levels were fasting

3. Methods (Page 6) – Please add details on which definition/criteria for Metabolic Syndrome was used for this study.

4. Methods (Page 6) – Please clarify how non-normally distributed variables were handled.

5. Methods (Page 6) – Please clarify if there was adjustment for multiple comparisons.

6. Discussion (Page 12) – While it may not be possible to fully explain why this study finding differed from other work in South Africa and Uganda, please expand upon whether the age, sex, antipsychotic treatments were similar to this study sample. A deeper examination of the differences in this population from South Africa (the source of most African studies on this subject) would establish why these studies should not be lumped together.

7. Discussion (Page 12) – Please clarify in the third paragraph discussing the obesity rates whether the comparison is with general population samples or with participants with psychotic disorders. Also, the varying rates across country are not fully explored – please add further discussion

8. Discussion (Page 12-13)– given the higher rates of obesity among women, it would be helpful to clarify if women and men had differences in demographic, medication, or other clinical factors that might be related.

9. Discussion (Page 14, 2nd paragraph) – please correct the typos in the second sentence “a 2.4% diabetes.”)

10. Discussion (Page 15) – the high rates of olanzapine usage are an important contributor to the metabolic health of these patients. Please clarify if the other African studies of metabolic health had similarly high rates of olanzapine or other SGAs with high metabolic risk.

11. Discussion (Page 15) – another key point for clarification in whether olanzapine does/does not contribute to metabolic health is whether there were demographic differences in those who were on olanzapine vs. another antipsychotic (age, sex, education level, age of onset) as these factors may confound this relationship.

12. Discussion – Please expand on the screening practices and treatment gaps within this particular treatment setting. Rates of monitoring of metabolic health for patients on olanzapine vary by treatment setting/provider – it would be helpful to not whether this is a common practice in this setting.

13. Discussion – Deinstitutionalization in Western countries impacted mental and physical health outcomes for patients with psychotic disorders. It would be interesting to discuss whether such practices have occurred in Kenya and whether that may influence the health outcomes in this population.

6. PLOS authors have the option to publish the peer review history of their article (what does this mean?). If published, this will include your full peer review and any attached files.

Reviewer #1: No

Reviewer #2: No

---

## [Author Response · Author response to Decision Letter 0]

27 Oct 2020

I have responded to all the concerns as indicated on the response to reviewers document attached.

I have updated the figure to PACE style .

---

## [Decision Letter · Decision Letter 1]

20 Nov 2020

PONE-D-20-18075R1

Prevalence and Correlates of Metabolic Syndrome and its Components in Adults with Psychotic Disorders in Eldoret, Kenya

PLOS ONE

Dear Dr. Kwobah,

Thank you for submitting your manuscript to PLOS ONE. After careful consideration, we feel that it has merit but does not fully meet PLOS ONE’s publication criteria as it currently stands. Therefore, we invite you to submit a revised version of the manuscript that addresses the points raised during the review process.

We look forward to receiving your revised manuscript.

Kind regards,

Alberto Milan

Academic Editor

PLOS ONE

Additional Editor Comments (if provided):

The paper has clearly improved, there are still several points to be reviewed by the authors

Reviewers' comments:

Reviewer's Responses to Questions

**Comments to the Author**

1. If the authors have adequately addressed your comments raised in a previous round of review and you feel that this manuscript is now acceptable for publication, you may indicate that here to bypass the “Comments to the Author” section, enter your conflict of interest statement in the “Confidential to Editor” section, and submit your "Accept" recommendation.

Reviewer #2: (No Response)

Reviewer #3: All comments have been addressed

2. Is the manuscript technically sound, and do the data support the conclusions?

Reviewer #2: Yes

Reviewer #3: Yes

3. Has the statistical analysis been performed appropriately and rigorously? 

Reviewer #2: Yes

Reviewer #3: Yes

4. Have the authors made all data underlying the findings in their manuscript fully available?

Reviewer #2: Yes

Reviewer #3: Yes

5. Is the manuscript presented in an intelligible fashion and written in standard English?

Reviewer #2: Yes

Reviewer #3: Yes

6. Review Comments to the Author

Reviewer #2: This is a study of metabolic syndrome among patients with psychosis and a non-psychiatric comparison group in Eldoret, Kenya. The study addresses an important problem in psychiatric treatment – the high rates of cardiometabolic disease and subsequent mortality among patients with psychosis. This is also one of the first studies from Kenya, as most studies are from South Africa, and showcase important clinical differences in low- and middle-income countries. The sample size is large (600 total), assessments and methodology are solid, and the discussion covers a number of relevant points regarding treatment gaps. The authors were able to put the study in context of other studies from Africa. I appreciate the author’s comprehensive response to our suggestions

1. Abstract – Results section: Please clarify that the marital status variable was widowed/separated/divorced (compared to never married and married groups). These were groupings show in Table 1.

2. Results – please note the age range of the participants in this study, to help clarify the relationship between older age and metabolic syndrome.

3. Results – please clarify whether the obesity findings in Table 4 were based on BMI or abdominal obesity. The BMI/abdominal obesity findings in the discussion should be mentioned in the results section as well.

4. Discussion – the finding that female gender was associated with decreased odds of metabolic syndrome but increased odds of obesity are puzzling and warrant further discussion.

5. Discussion- the discrepancy in gender findings on metabolic syndrome from the current study and South African study are also puzzling. Were there age/other demographic factor differences between the two cohorts? What were the findings regarding obesity in the two groups?

6. Discussion – additional interpretation/synthesis would be helpful to understand why the present study would differ from the Western Kenya study. Are the two groups of participants comparable in terms of demographic features or treatments?

7. Discussion – the low numbers of participants on agents other than olanzapine make it difficult to truly link olanzapine use to metabolic problems. It may be helpful to emphasize the finding that dosage of olanzapine was not linked to metabolic syndrome or other cardiometabolic abnormalities.

8. Discussion – please clarify what is indicated by “human resources for mental healthcare” – would those be increasing the staffing of mental health clinics or other types of support?

Reviewer #3: This article analyses the correlation of metabolic syndrome and psychotic disorders in adults with a diagnosis of schizophrenia, schizoaffective disorder or bipolar mood disorder, compared to healthy individuals iin Kenya.

This paper which focuses on a low resource context, which is often overlooked. It shows a significant increase of certain metabolic syndrome parameters in patients vs. controls, such as HDL levels, waist circumference and TG.

It is an observational study with definite limitations, however it seems reasonable and well written

7. PLOS authors have the option to publish the peer review history of their article (what does this mean?). If published, this will include your full peer review and any attached files.

Reviewer #2: No

Reviewer #3: No

---

## [Author Response · Author response to Decision Letter 1]

25 Nov 2020

I have revised the manuscript and provided a clean manuscript, a revised manuscript with track changes and a cover letter detailing the response to reviewers

---

## [Decision Letter · Decision Letter 2]

22 Dec 2020

Prevalence and Correlates of Metabolic Syndrome and its Components in Adults with Psychotic Disorders in Eldoret, Kenya

PONE-D-20-18075R2

Dear Dr. Kwobah,

We’re pleased to inform you that your manuscript has been judged scientifically suitable for publication and will be formally accepted for publication once it meets all outstanding technical requirements.

Kind regards,

Alberto Milan

Academic Editor

PLOS ONE

Additional Editor Comments (optional):

Reviewers' comments:

Reviewer's Responses to Questions

**Comments to the Author**

1. If the authors have adequately addressed your comments raised in a previous round of review and you feel that this manuscript is now acceptable for publication, you may indicate that here to bypass the “Comments to the Author” section, enter your conflict of interest statement in the “Confidential to Editor” section, and submit your "Accept" recommendation.

Reviewer #2: All comments have been addressed

Reviewer #3: All comments have been addressed

2. Is the manuscript technically sound, and do the data support the conclusions?

Reviewer #2: Yes

Reviewer #3: Yes

3. Has the statistical analysis been performed appropriately and rigorously? 

Reviewer #2: Yes

Reviewer #3: Yes

4. Have the authors made all data underlying the findings in their manuscript fully available?

Reviewer #2: Yes

Reviewer #3: Yes

5. Is the manuscript presented in an intelligible fashion and written in standard English?

Reviewer #2: Yes

Reviewer #3: Yes

6. Review Comments to the Author

Reviewer #2: This is a study of metabolic syndrome among patients with psychosis and a non-psychiatric comparison group in Eldoret, Kenya. The study addresses an important problem in psychiatric treatment – the high rates of cardiometabolic disease and subsequent mortality among patients with psychosis. This is also one of the first studies from Kenya, as most studies are from South Africa, and showcase important clinical differences in low- and middle-income countries. The sample size is large (600 total), assessments and methodology are solid, and the discussion covers a number of relevant points regarding treatment gaps. The authors were able to put the study in context of other studies from Africa. I appreciate the author’s comprehensive response to our suggestions and believe the resultant manuscript is a clear and meaningful contribution to the literature.

Reviewer #3: no further comments the authors addressed all the comments raised; no further comments as previous one

7. PLOS authors have the option to publish the peer review history of their article (what does this mean?). If published, this will include your full peer review and any attached files.

Reviewer #2: No

Reviewer #3: No

---

## [Editor Report · Acceptance letter]

29 Dec 2020

PONE-D-20-18075R2 

Prevalence and Correlates of Metabolic Syndrome and its Components in Adults with Psychotic Disorders in Eldoret, Kenya 

Dear Dr. Kwobah:

I'm pleased to inform you that your manuscript has been deemed suitable for publication in PLOS ONE. Congratulations! Your manuscript is now with our production department. 

Kind regards, 

on behalf of

Dr. Alberto Milan 

Academic Editor

PLOS ONE